

# Slc26a1 is not essential for spermatogenesis and male fertility in mice

Zhixiang Meng[1],[*], Yu Qiao[2],[*], Jiajia Xue[1], Tiantian Wu[3], Wenxin Gao[3], Xiaoyan Huang[3], Jinxing Lv[1], Mingxi Liu[4] and Cong Shen[5]

[1] Dushu Lake Hospital Affiliated to Soochow University, Center for Reproduction, Suzhou Dushu Lake Hospital, Suzhou, Jiangsu, China
[2] The Affiliated Huai'an No.1 People's Hospital of Nanjing Medical University, Center for Reproduction, Huai'an, Jiang Su, China
[3] Department of Histology and Embryology, School of Basic Medical Sciences, Nanjing Medical University, State Key Laboratory of Reproductive Medicine, Nanjing, Jiangsu, China
[4] State Key Laboratory of Reproductive Medicine and Offspring Health, The Affiliated Taizhou People's Hospital of Nanjing Medical University, Taizhou School of Clinical Medicine, Nanjing Medical University, Jiangsu, China
[5] The Affiliated Suzhou Hospital of Nanjing Medical University, Gusu School, Nanjing Medical University, State Key Laboratory of Reproductive Medicine, Center for Reproduction and Genetics, Suzhou Municipal Hospital, Suzhou, Jiangsu, China
[*] These authors contributed equally to this work.

## ABSTRACT

Thousands of genes are expressed in the testis of mice. However, the details about their roles during spermatogenesis have not been well-clarified for most genes. The purpose of this study was to examine the effect of Slc26a1 deficiency on mouse spermatogenesis and male fertility. Slc26a1-knockout (KO) mice were generated using CRISPR/Cas9 technology on C57BL/6J background. We found no obvious differences between Slc26a1-KO and Slc26a1-WT mice in fertility tests, testicular weight, sperm concentrations, or morphology. Histological analysis found that Slc26a1-KO mouse testes had normal germ cell types and mature sperm. These findings indicated that Slc26a1 was dispensable for male fertility in mice. Our results may save time and resources by allowing other researchers to focus on genes that are more meaningful for fertility studies. We also found that mRNAs of two Slc26a family members (Slc26a5 and Slc26a11) were expressed on higher mean levels in Slc26a1-KO total mouse testes, compared to Slc26a1-WT mice. This effect was not found in mouse GC-1 and GC-2 germ cell lines with the Slc26a1 gene transiently knocked down. This result may indicate that a gene compensation phenomenon was present in the testes of Slc26a1-KO mice.

## INTRODUCTION

The solute carrier 26A (SLC26A) family member is a highly conserved membrane protein that regulates transport of various anions (e.g., chloride ($Cl^-$), bicarbonate ($HCO_3^-$), sulfate ($SO_4^{2-}$), iodide ($I^-$), formate ($HCOO^-$), and oxalate ($C_2O_4^{2-}$)) across the plasma membrane of epithelial cells, regulating the composition and pH of body secretions (El Khouri et al., 2018). SLC26A-related proteins are present in a variety of organisms, including bacteria,

Corresponding authors
Mingxi Liu, mingxi.liu@njmu.edu.cn
Cong Shen, 344128944@qq.com

yeast, algae, plants (SulP/Sultr proteins), and non-mammalian vertebrates (*Mount & Romero, 2004*). The *SLC26A* family members are encoded 11 identified genes, but *SLC26A10* is a pseudogene in humans (*Alper & Sharma, 2013*). In humans, *SLC26A* family members are located in different tissues and have important roles in maintenance of ion balance and pH values (*Bernardino et al., 2019*). Mutations in these genes lead to unique clinical diseases related to their specific distribution: kidney stones (*SLC26A1*), maldevelopment (*SLC26A2*), congenital chloride diarrhea (*SLC26A3*), Pendred syndrome and goiter (*SLC26A4*), deafness (*SLC26A5*), and male infertility (*SLC26A3, SLC26A8*) (*Dawson & Markovich, 2005*; *El Khouri & Touré, 2014*; *Everett & Green, 1999*).

The effects of the *SLC26A* family members on male fertility remain poorly described. *SLC26A8* (*i.e.*, testicular anion transporter 1, TAT1) was first identified in 2001 and specifically expressed in spermatocytes and spermatids (*Lohi et al., 2002*; *Toure et al., 2001*). Generation of *Slc26a8* knockout (KO) mice results in male infertility due to decreased sperm motility, but mouse viability is not affected. This result indicates that *SLC26A8* is a key factor in sperm differentiation and sperm motility (*Touré et al., 2007*). In addition to induction of congenital chloride-deficient diarrhea (CLD), the *SLC26A3* gene is expressed in the male reproductive tract and sperm cells (*Chávez et al., 2012*). Previous studies that analyzed male reproductive parameters and functions in *Slc26a3*-KO mice found that *Slc26a3* deletion is associated with severe lesions and abnormal cell structure in the epididymis and defects in sperm number, morphology, and function; these effects jointly impair male fertility (*El Khouri et al., 2018*). *SLC26A4* and *SLC26A7* are present in the testis, epididymis, seminiferous duct, and ejaculatory duct (*Pierucci-Alves, Akoyev & Schultz, 2015*). Taken together, these study findings indicate that it is necessary to further understand the effects of the *SLC26A* families on male fertility.

*SLC26A1/SAT1* cDNA was first cloned from rat liver tissue; it is a $Na^+$-independent transporter (*Bissig et al., 1994*). Human *SLC26A1* shows moderate cross-species conservation, with 78% homology to rat and 77% homology to mouse cDNA (*Regeer, Lee & Markovich, 2003*). Mice with a *Slc26a1* deficiency exhibit urolithiasis, hyperoxaluria, hyperoxalemia, hyposulfatemia, hypersulfaturia, and increased susceptibility to hepatotoxicity (*Dawson et al., 2010*). *SLC26A1* is expressed in the testis, as well as in the kidney and liver (*Yin et al., 2017*). However, no studies on the role of *Slc26a1* in mouse fertility and no human infertility caused by *SLC26A1* mutation have been reported. In this study, we examined the role of *Slc26a1* in spermatogenesis and male fertility. We generated *Slc26a1*-deficient mice, investigated the effects of *Slc26a1*-deficiency on the murine testis and epididymis using histological and immunohistochemical staining, and assessed quality of spermatozoa using a computer-assisted sperm analyzer. We found that male mice with *Slc26a1*-KO displayed normal fertility parameters. These results suggested that the effect of *Slc26a1* deficiency on male mouse fertility is dispensable. We speculate that these results can be explained by genetic redundancy.

## MATERIALS AND METHODS

### Generation of *Slc26a1*⁻/⁻ mice using CRISPR/Cas9 technology

C57BL/6J healthy mice were purchased from Cyagen Biosciences Inc and raised at the Experimental Animal Center of Nanjing Medical University under conditions of 30–70% humidity and 26 °C. The mice were allowed free access to food and water throughout the experiment. The food contains 20% protein and 4% fat and was provided Xietong Pharmaceutical Bio-engineering (Nanjing, China). All the cages were exactly of the same size and material. Animal house was specific pathogen-free (*Murray et al., 2021*) and 12 h light-12 h darkness-cycles. At the end of the study, the mice were anesthetized with carbon dioxide.

This study was approved by the Animal Ethics and Welfare Committee of Nanjing Medical University (No. IACUC-2004020) and was conducted in accordance with the Guide for the Care and Use of Laboratory Animals. *Slc26a1* KO mice were generated using CRISPR/Cas9 genome editing technology, as previously described (*Shen et al., 2014*; *Wang et al., 2013*). Single guide (sg)RNA target sequences for *Slc26a1* are as follows: 5′-CATGGTGGTCCACACATGGT-3′. The combination of sgRNA and Cas9mRNA was microinjected into the fertilized eggs of C57BL/6J mice and then immediately transplanted into the oviduct of pseudopregnant females of the same strain. The mutant mice have been registered in the MGI database as *Slc26a1*<em1Njrml> (MGI:7529731).

### Genotype identification

*Slc26a1*⁻/⁻ mutation was identified by Sanger sequencing. Polymerase chain reaction (PCR) and Sanger sequencing were performed using the forward primer, 5′-GGCTGGG CTTCGTGTCTACCTA-3′, and the reverse primer, 5′-GCTCTTGGTTGGCACTGACAG A-3′.

### Fertility test

*Slc26a1* (WT/KO) adult males were respectively mated with WT females at a ratio of 1:2 for 3 months. The mice were raised at 26 °C. Female mice, whether pregnant or not, were replaced once a week. The numbers of litters and born mice per litter were recorded. The genetic background of WT controls was C57BL/6J (from the same batch as KO mice but not littermates).

### Cell culture and transfection

Mouse germ cell lines GC-1spg (GC-1) and GC-2(spd)ts (GC-2) were purchased from the American Type Culture Collection (Manassas, VA, USA). The cells were cultured in DMEM medium containing 10% fetal bovine serum (FBS; Gibco, Grand Island, NY, USA) under the conditions of 37 °C and 5% $CO_2$. *SLC26A1* and negative control small interfering RNAs (si-NC) were synthesized by GenePharma (Suzhou, China). GC-1 and GC-2 cells were transfected with Lipofectamine 2000 (Invitrogen, Carlsbad, CA, USA). At 6 h post-transfection, cells were transferred to culture medium; at 48 h post-transfection, cells were collected for analysis. The sequences of the small interfering RNAs (siRNAs) were:

si-NC:5′-CACUCAAGAUUGUCAGCAA-3′;
si-*Slc26a1* #1:5′-GGAAUACCUAGCAGGUGAU-3′;
si-*Slc26a1* #2:5′-GCCAAUACCCACAGAGUUA-3′

## Cell viability and migration analysis

After transfection with siRNA, GC-1 and GC-2 cells were seeded in 96-well plates at a density of $4 \times 10^3$ cells/well. Cell viability was evaluated using a cell counting kit-8 (CCK-8; Beyotime Institute of Biotechnology, Nantong, China), as previously described (*Chen et al., 2022a*; *Xue et al., 2022*; *Yu et al., 2022*). In the colony formation assay, 1,000 transfected cells were placed in 6-well plates; 2 mL DMEM containing 10% FBS was added to each well and changed after 6 d. After 2 weeks, the cells were fixed with methanol and stained with 0.1% crystal violet (Beyotime, Jiangsu, China) for 30 min, and the visible proliferating clumps were counted under a microscope (Carl Zeiss, Oberkochen, Germany).

In the migration assay, 300 µL serum-free DMEM was added to the top of a culture chamber with an 8-mm membrane (14831; Corning, Corning, NY, USA), and the cells were then added; 700 µL DMEM containing 10% FBS was added to the lower part of the chamber. After 24–48 h, the cells below the membrane were fixed with 4% paraformaldehyde and stained with 0.1% crystal violet. Finally, photographs were taken under a microscope (Carl Zeiss, Oberkochen, Germany).

## Western blot

The proteins of transfected GC-1 and GC-2 cells were extracted using RIPA buffer (RIPA, Beyotime, Jiangsu, China) and denatured at 100 °C. The denatured proteins were then separated on 10% sodium dodecyl sulfate-polyacrylamide gels, transferred to a polyvinylidene difluoride membrane, and sealed with 5% skim milk at room temperature for 1 h. The primary antibodies were anti-*SLC26A1* (1:1,000, NBP1-84897; Novus, St. Charles, MO, USA) and anti-tubulin antibody (1:1,000, MA1-91878, Thermo Fisher Science, Waltham, MA, USA). The membranes were then incubated with secondary antibody (1:3,000, A0208, Beyotime, Jiangsu, China) for 1 h at room temperature. Finally, a quantitative analysis was performed using a SuperSignal West Femto chemiluminescence substrate detection system (Thermo Fisher Scientific, Waltham, MA, USA).

## RNA extraction and reverse transcription-quantitative PCR (RT-qPCR)

Total RNA was extracted from testicular tissue and transfected GC-1 and GC-2 cells using TRIZOL™ reagent (Vazyme, Nanjing, China) according to the manufacturer's instructions. The RNA was then reverse transcribed into complementary cDNA by using a PrimeScript RT reagent Kit (Vazyme, Nanjing, China) and quantified using a real-time PCR system (Applied Biosystems, Foster City, CA, USA). The reaction system was set up as follows: SYBR Green Mix (10 µL), forward primer (0.4 µL), reverse primer (0.4 µL), cDNA template (1.0 µL), ddH2O (8.2 µL), resulting in a total volume of 20 µL. 18S rRNA was used as an internal control. Target gene expression was calculated using the $2^{-\Delta\Delta CT}$ method: $\Delta\Delta CT = (CT_{Target} - CT_{18srRNA})_{Sample} - (CT_{Target} - CT_{18srRNA})_{Control}$. The primers used are summarized in Table S1.

### Histology and sperm phenotyping

Testes and epididymides from *Slc26a1*-WT and *Slc26a1*-KO mice aged 8–12 weeks were immediately fixed in modified Davidson's solution for 48 h, dehydrated in a series of graded ethanol solutions, embedded in paraffin, and cut into sections (thickness, 4 μm). The sections were rehydrated and stained for histological analysis using hematoxylin-eosin (H&E) or periodic acid Schiff (PAS) stain, as previously described (*Shen et al., 2021*; *Wu et al., 2022*; *Yu et al., 2021*). For sperm malformation analysis, we selected and quantified the head and tail deformities of sperm, respectively. Then, we performed computer-assisted sperm analysis (CASA). Cauda epididymides were suspended in human tubal fluid culture medium (InVitroCare, Inc., Frederick, MD, USA), incubated at atmospheric pressure and 37 °C for 10 min, and analyzed for semen quality using a Ceros™II sperm analysis system (Hamilton Thorne, Beverly, MA, USA).

### Immunofluorescence

Immunofluorescence staining of mouse testis was performed as previously described (*Gao et al., 2020*; *Zhao et al., 2019*). The paraffin sections were boiled in 10 mM citrate buffer (pH 6.0), soaked in 3% hydrogen peroxide, blocked with bovine serum albumin, and incubated with primary antibody (Table S2) at 4 °C. Subsequently, the slides were rinsed three times with phosphate-buffered saline, then incubated with Alexa-Fluor secondary antibody (Thermo Fisher Scientific, Waltham, USA) at 37 °C and stained with 4′,6-diaminodiphenylindole. Fluorescent staining images of all sections were obtained under a confocal microscope (Zeiss, Oberkochen, Germany). Images of 50 tubules per male were used for quantification.

### Terminal deoxynucleotidyl transferase-dUTP nick-end labeling (TUNEL) assay

Apoptosis was detected using the TUNEL assay kit (Vazyme, Nanjing, Jiangsu, China), according to the manufacturer's instructions. Paraffin sections were rehydrated and treated with proteinase K, then reacted with TUNEL labeling mix buffer at 37 °C. Images were obtained under a confocal microscope (Zeiss, Oberkochen, Germany). Fifty tubules per male were analyzed.

### Statistical analysis

All relevant statistical analyses were performed in triplicate. Unpaired student's t-tests or one-way ANOVA were used to verify differences between *Slc26a1*-WT and *Slc26a1*-KO mice. Data are the mean ± Standard Deviation (SD). A *P*-value < 0.05 was considered to indicate a significant difference.

## RESULTS

### Generation of *Slc26a1*-KO mice

To investigate the effect of *SLC26A1* on male fertility, CRISPR/Cas9 gene editing technology was used to introduce a mutation consisting of a 10 bp deletion in exon 3 of *Slc26a1* (Fig. 1A), which is predicted to lead to frame-shift mutations. This mutation

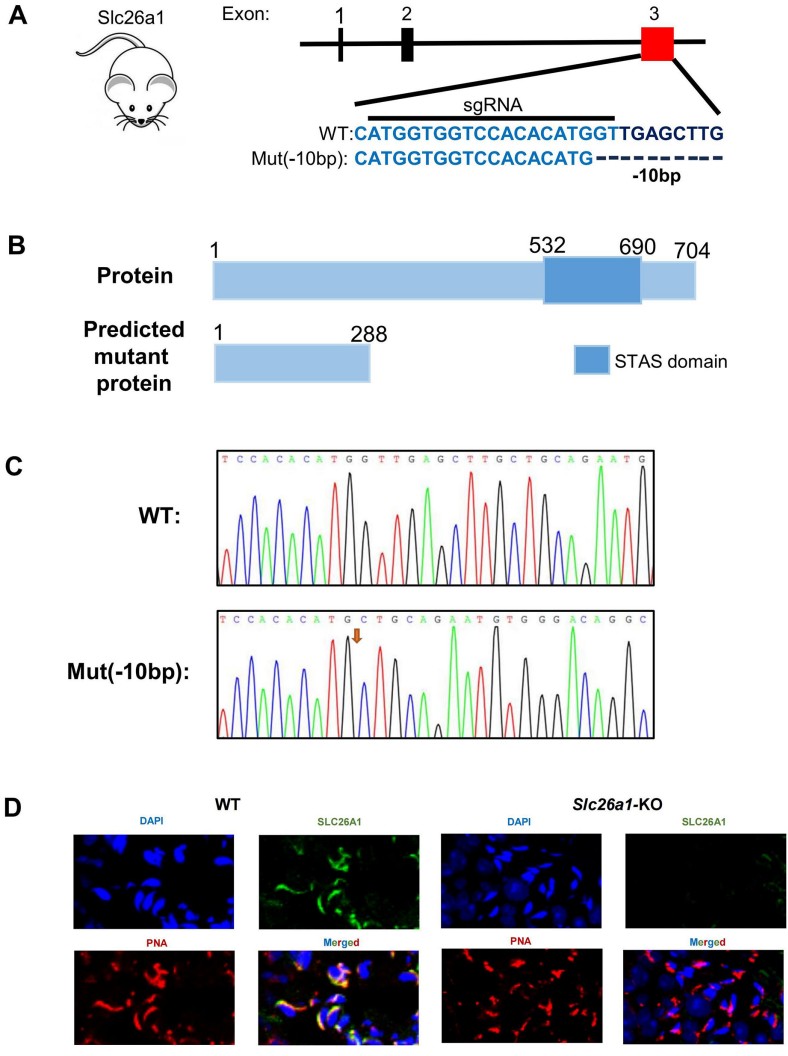

**Figure 1 Generation of *Slc26a1*-KO mice.** (A) CRISPR/Cas9-mediated gene editing strategy for *Slc26a1*. (B) Predictive analysis of mutant proteins in *Slc26a1*-KO mice. (C) Sanger sequencing of *Slc26a1-WT* and *Slc26a1-KO* mice. (D) Immunostaining of *SLC26A1* and PNA in the sperm of WT and *Slc26a1*-KO mice. PNA served as an acrosome marker. The *SLC26A1* antibody recognition site is at the C-terminal, the antibody recognizes only the WT form, not the mutant protein. Scale bar, 20 μm. *SLC26A1*, solute carrier 26A; WT, wild type; DAPI, 4′,6-diamidino-2-phenylindole; PNA, peanut agglutinin.

causes a complete deletion of the Sulfate Transport Anti-Sigma antagonist (STAS) domain (Fig. 1B). PCR and Sanger sequencing were used to confirm the changes (Fig. 1C). Immunofluorescence staining of sperm showed that *SLC26A1* was specific localized in the acrosome of spermatids of WT mice, but not in the *Slc26a1*-KO spermatids (Fig. 1D). Localization of *Slc26a1* in mice suggested that *Slc26a1* may affect acrosome during spermatogenesis.

### Slc26a1 KO mice are fertile

We subsequently performed fertility tests to study the effects of *Slc26a1* on mouse fertility. The pups and litters of the experimental and control groups were counted. The results suggested that the difference in fertility between the *Slc26a1*-WT and *Slc26a1*-KO mice was not statistically significant (Fig. 2A). We also examined testicular morphology and testicular/body weight ratios of *Slc26a1*-KO male mice; compared with the control group, the difference was not statistically significant (Figs. 2B and 2C). The CASA results suggested that compared with WT mice, *Slc26a1*-KO mice had normal sperm concentrations, and percentage of sperm with motility and with progressive motility (Figs. 2D–2F). H&E staining of sperm found that the abnormal sperm of *Slc26a1*-KO mice was not significantly different from that of *Slc26a1*-W mice (Figs. 2G–2I). Therefore, *Slc26a1* was not essential for mouse fertility.

### Spermatogenesis is normal in *Slc26a1*-KO mice

H&E staining results for testes suggested that *Slc26a1*-KO mice had intact seminiferous tubules and spermatogenic cells at all stages (Fig. 3A). H&E staining of the epididymis found that there were no significant histological differences between the epididymides of *Slc26a1*-WT and *Slc26a1*-KO mice and that they were filled with sperm in both groups of animals (Figs. 3B and 3C).

Different spermatogenic cells are arranged in the seminiferous tubules according to special cell connections. Spermatogenesis can be divided into the three processes of spermatogonial stem cell (SSC) proliferation and differentiation, spermatocyte meiosis, and spermiogenesis. To examine the differences in spermatogenesis between *Slc26a1*-WT and *Slc26a1*-KO mice, we first analyzed the nuclei and acrosomes of the spermatogenic cells in seminiferous tubules using PAS staining. The results indicated there were no obvious morphological differences (Fig. 4A). We quantitatively analyzed spermatids in spermatogenic tubules by H&E-stained mouse testicular sections (Fig. 4B), the results suggest that there is no significant difference in the numbers of spermatids between WT and KO mice.

We also used lin-28 homolog (LIN28), SRY-box 9 (SOX9), hydroxy-delta-5-steroid dehydrogenase, 3 beta- and steroid delta-isomerase 1(HSD-3β), TUNEL, and H2AX variant histone (γ-H2AX) immunostaining. These five marker genes are respectively located in SSCs (Figs. 4C and 4D), Sertoli cells (Figs. 4E and 4F), Leydig cells (Figs. 4G and 4H), apoptotic cells (Figs. 4I–4K), and spermatocytes (Figs. 4L and 4M). Then we quantified the number of positive cells in the testicular sections; there were no significant between-group differences. Taken together, these results suggested that *Slc26a1* was not essential for spermatogenesis in mice.

### Possible functional compensation from paralogs in *Slc26a1*-KO mice

We extracted total RNA from the testes of *Slc26a1*-WT and *Slc26a1*-KO mice. Relative transcript levels of eleven *Slc26a* family members were detected using RT-qPCR. The mean expression of *Slc26a5* and *Slc26a11* in the testes of *Slc26a1*-KO mice was higher than that

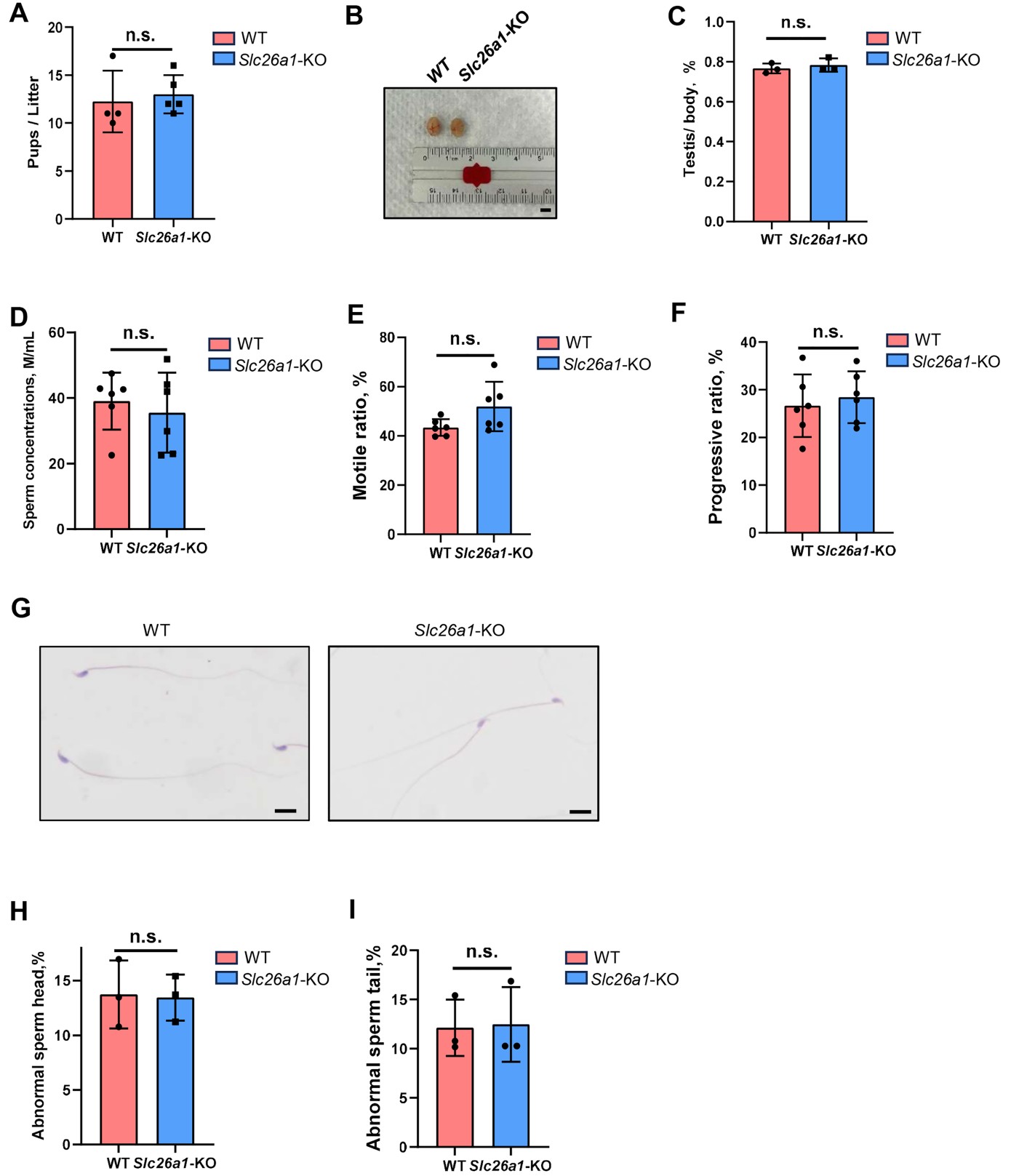

**Figure 2 Normal fertility in *Slc26a1*-KO mice.** (A) Fertility test of *Slc26a1*-WT and *Slc26a1*-KO mice; to five males (n) per group *P* > 0.05. (B) Representative examples of testicular morphology of 8-week-old *Slc26a1*-WT and *Slc26a1*-KO mice. Scale bar = 5 mm. (C) Testis/body weight ratio

**Figure 2** (continued)

of the control group and the experimental group; *n* = 6 per group, *P* > 0.05. (D–F) Sperm concentrations (D), motility (E), and progressive motility (F) in *Slc26a1*-WT and *Slc26a1*-KO mice. M, millions; *n* = 6 per group, *P* > 0.05. (G) Examples of H&E staining of sperm from *Slc26a1*-WT and *Slc26a1*-KO mice. Scale bar, 20 μm. (H) Percentages of abnormal sperm head in *Slc26a1*-WT and *Slc26a1*-KO mice; *n* = 3 per group, 500 sperm cells were counted in each group. *P* > 0.05. (I) Percentages of abnormal sperm tail in *Slc26a1*-WT and *Slc26a1*-KO mice; *n* = 3 per group, 500 sperm cells were counted in each group. *P* > 0.05. 

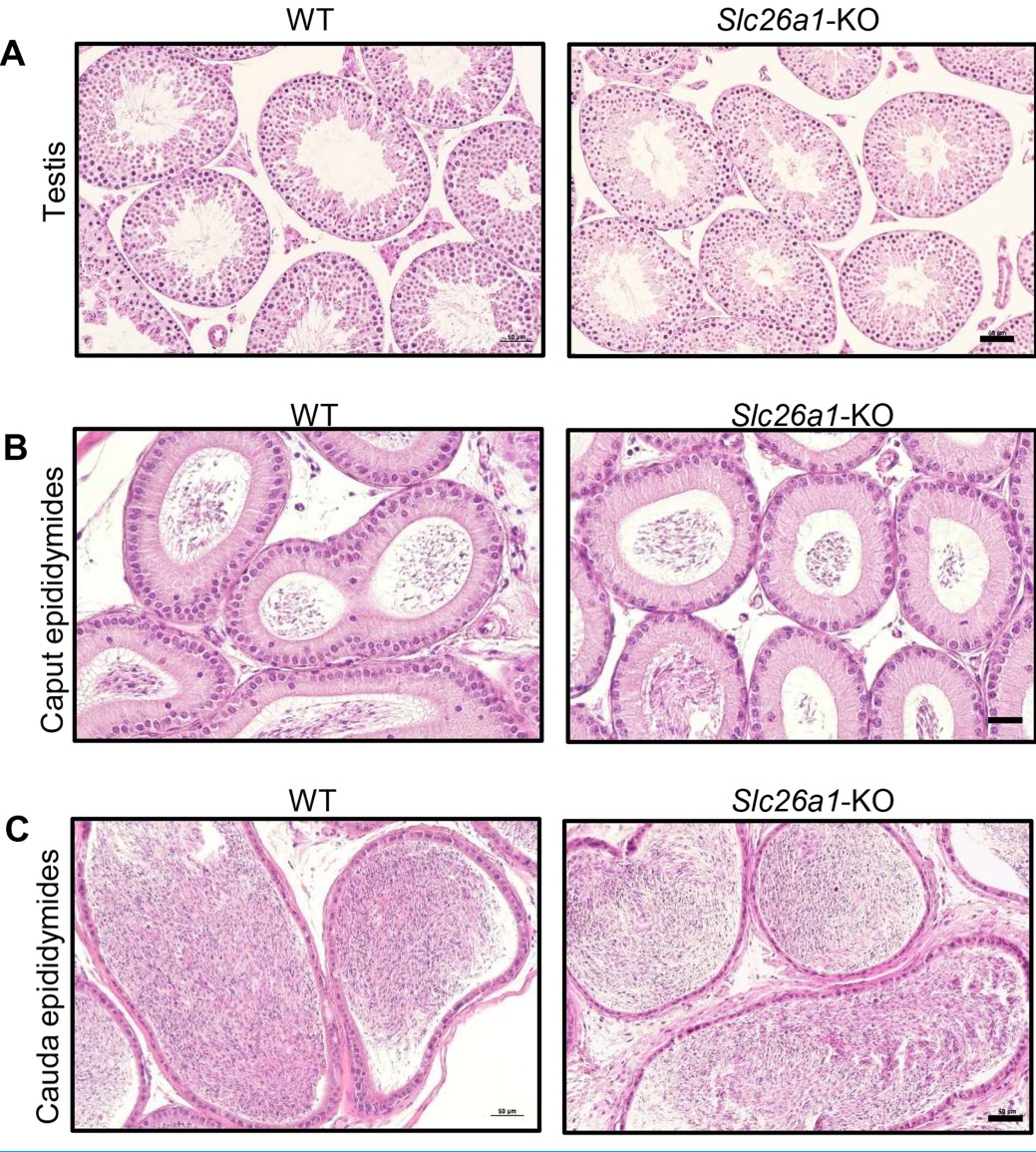

**Figure 3 Histological analysis of testes and epididymides from 8-week-old mice.** (A) Representative H&E staining of testicular sections from *Slc26a1*-WT and *Slc26a1*-KO mice, scale bar = 50 μm. (B) Examples of H&E staining of caput epididymal sections from *Slc26a1*-WT and *Slc26a1*-KO mice, scale bar = 50 μm. (C) H&E staining of cauda epididymis sections from *Slc26a1*-WT and *Slc26a1*-KO mice, scale bar = 50 μm. 

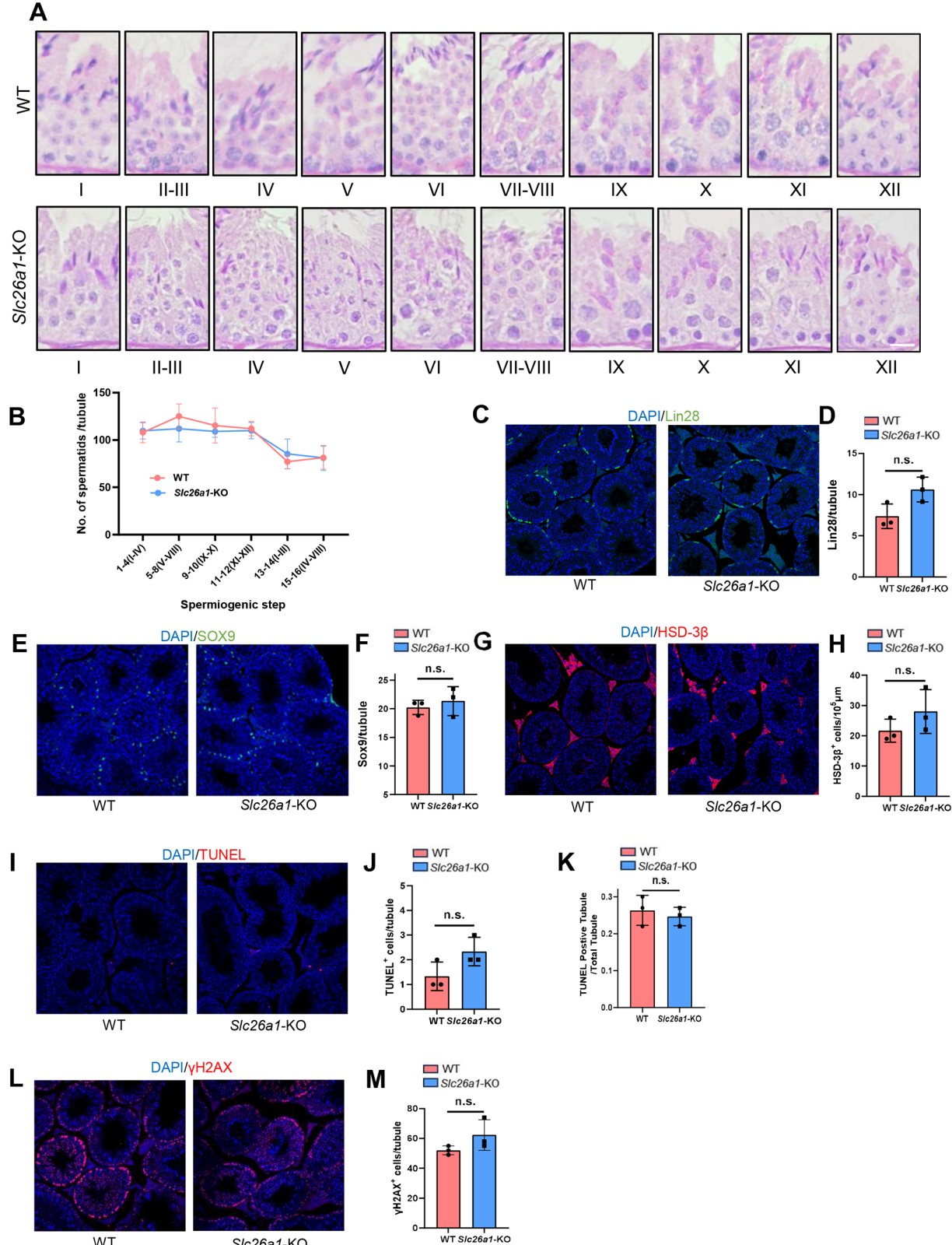

**Figure 4 Normal spermatogenesis in *Slc26a1*-KO mice.** (A) Representative images of PAS staining of spermatogenic cells (epithelial stages I–XII) from *Slc26a1*-WT and *Slc26a1*-KO mice, scale bar = 20 μm. (B) The number of spermatids in each spermatogenic tubule were counted in

**Figure 4 (continued)**
H&E-stained mouse testicular sections. (C) Representative immunostaining of LIN28 in testicular sections of 8-week-old *Slc26a1*-WT and *Slc26a1*-KO mice. Scale bar = 50 μm. (D) Comparative quantification of LIN28-positive cells (spermatogonial stem cells, SSCs) per tubule (based on (C), *n* = 3 per group, *P* > 0.05. (E) Immunostaining of SOX9 (served as a Sertoli cell marker) in 8-week-old *Slc26a1*-WT and *Slc26a1*-KO mice. Scale bar = 50 μm. (F) Quantification of (E), *n* = 3 per group, *P* > 0.05. (G) Immunostaining of 3β-HSD (served as a Leydig cell marker) in 8-week-old *Slc26a1*-WT and *Slc26a1*-KO mice. Scale bar = 50 μm. (H) Quantification of (G), *n* = 3 per group, *P* > 0.05. (I) TUNEL assay of testes from *Slc26a1*-WT and *Slc26a1*-KO mice, scale bar = 50 μm. (J) Quantification of apoptotic cells (based on I), *n* = 3 per group, *P* > 0.05. (K) Percentage of TUNEL-positive tubules (apoptotic cells) in total spermatogenic tubules, 50 tubules were counted in each group. (L) Immunostaining of γH2AX (served as a spermaticyte marker) in testes of *Slc26a1*-WT and *Slc26a1*-KO 8-week-old mice, scale bar = 50 μm. (M) Quantification of primary spermatocytes (base on L), *n* = 3 per group, *P* > 0.05. Scale bar = 50 μm.               

of *Slc26a1*-WT mice (Fig. 5A). These results may suggest that paralog-associated functional compensation was present in the testes of *Slc26a1*-KO mice.

To test whether transient knockdown of the *Slc26a1* gene also had a similar effect, we transiently transfected GC-1 and GC-2 cells with si-*Slc26a1* #1 and #2; western blot analysis revealed reduced expression of *SLC26A1* (Figs. 5B and 5C). We also extracted total RNA from GC-1 and GC-2 cells and performed RT-qPCR. The PCR results indicated there was no significant difference in relative transcription levels of Slc26a family members between *Slc26a1*-WT and *Slc26a1*-KO mice (Figs. 5D and 5E).

Since there was no compensation in these two cell lines, we assessed whether the knockdown of *Slc26a1* affects their phenotype. The results of the CCK-8 and colony formation assays revealed that proliferation of GC-1 and GC-2 cells decreased with downregulation of *Slc26a1* expression (Figs. 5F–5J). Transwell assays further found that downregulation of *Slc26a1* inhibited cell migration (Figs. 5K–5M).

Taken together, these results indicated that the deletion of *Slc26a1* could be compensated for by other *Slc26a* family members in some testicular cells, but transient gene knockdown in GC-1 and GC-2 cells had no such effect.

## DISCUSSION

As anion transporters, *SLC26A* family members have important roles in pH homeostasis, sperm maturation, and capacitation in the male reproductive tract. Previous studies found that the SLC26A family members (*e.g.*, SLC26A3 and SLC26A8 proteins) are involved in the regulation of germ cells (*Touré, 2019*). However, the role of *Slc26a1* in mouse spermatogenesis and fertility has remained unknown. In this study, we examined *SLC26A1* and obtained *Slc26a1*-KO mice using CRISPR/Cas9 technology. We found no significant differences in spermatogenesis or fertility of *Slc26a1*-KO mice. This result indicated that *Slc26a1* was dispensable for mouse fertility. However, in the *Slc26a1* knockdown model, proliferation and migration ability of mouse GC-1 and GC-2 cells were decreased. Therefore, there were phenotypic differences between the KD and KO models. Similar to the findings of previous studies on *Rai14* (*Wu et al., 2021*) and *Fank1* (*Zhang et al., 2019*), this difference might have been due to functional compensation of homologs in the KO model in some germ cells.

Genetic robustness refers to the ability of living organisms to maintain viability and adaptability under conditions of genetic variation (including disturbance; *Waddington,*

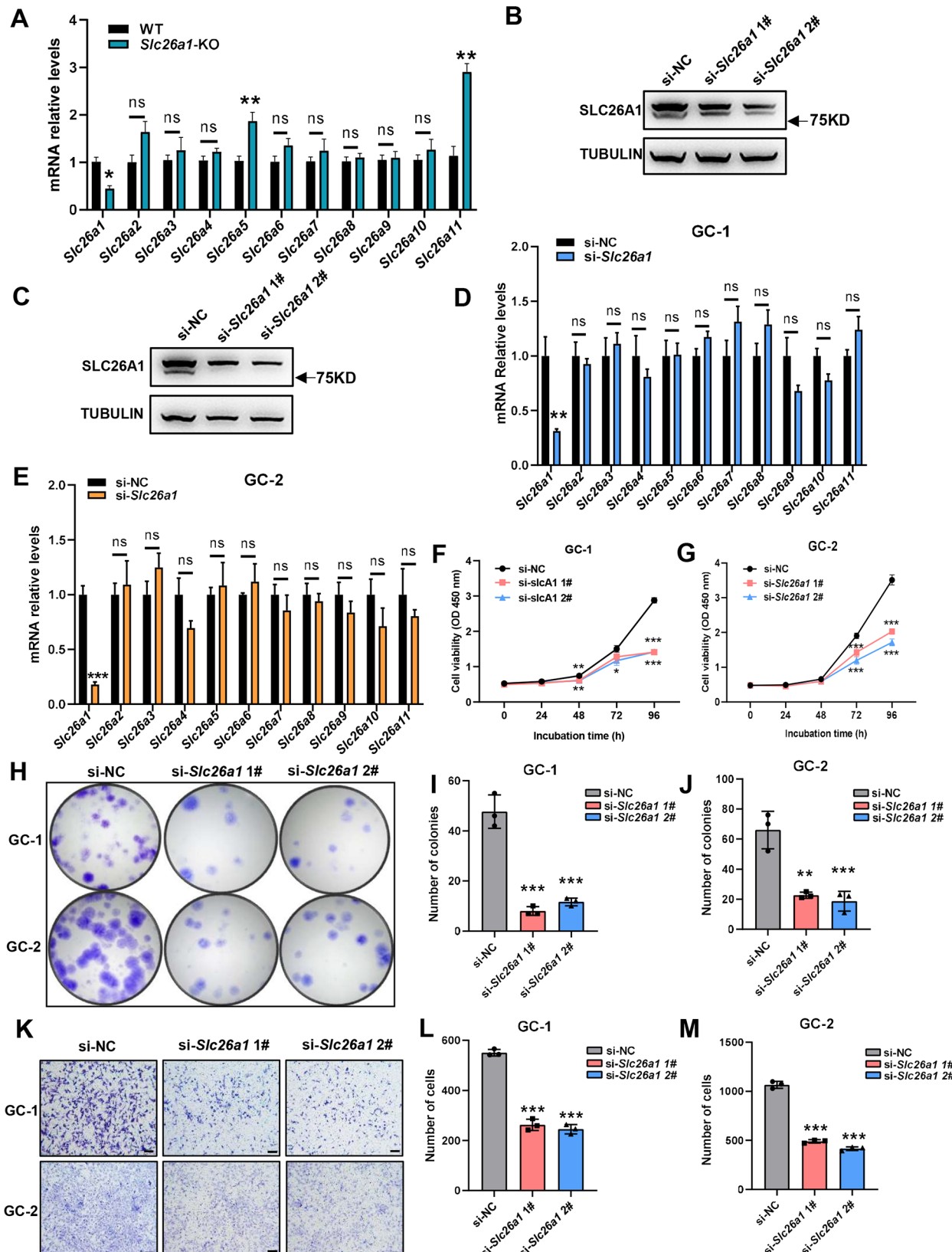

**Figure 5 Putative compensation in *Slc26a1*-KO mice.** (A) Relative mRNA levels of eleven *Slc26a* family genes in total testes of *Slc26a1*-WT and *Slc26a1*-KO mice, analyzed using RT-qPCR. $^*P < 0.05$, six males per group. (B and C) Western blot analysis revealed that *Slc26a1* was knocked down

**Figure 5 (continued)**
in *Slc26a1*-specific-siRNA-treated GC-1 (B) and GC-2 (C) cells; tubulin was used as an internal reference. (D and E) Relative mRNA levels of Slc26a family genes in GC-1 (D) and GC-2 (E) cells, using RT-qPCR. $*P < 0.05$, $**P < 0.01$, $***P < 0.001$. (F and G) Viability of GC-1 (F) and GC-2 (G) cells assessed using CCK-8 assay 48 h after transfection, $n = 3$ per group. (H–J) Colony formation assay performed to determine proliferation capacity of GC-1 (I) and GC-2 (J) cells 48 h after transfection, $n = 3$ per group. (K–M) Migration ability of GC-1 (L) and GC-2 (M) cells assessed using transwell assay 48 h after transfection.               

*1959*). When there are small differences in genetic composition or environmental conditions, organisms need buffer systems to ensure similar developmental outcomes; this process is known as robustness (*Waddington, 1959*). As previously reported (*White et al., 2013*), this genetic robustness may come from a redundant gene. Gene loss can be compensated for by other genes with overlapping functions and the same gene expression patterns (*Bouché & Bouchez, 2001*; *Kok et al., 2015*; *White et al., 2013*).

We speculate that the minimal effect of *Slc26a1* on mouse fertility was due to functional redundancy. Fertility is an important function, and there may be redundant genes with overlapping functions to ensure stability of individual fertility. *Asb-1, 2, 3, 4, 5, 7, 8, 9, 11, 14, 15*, *17*, and *18* genes are expressed at significantly higher levels in *Asb12*-KO mice than in *Asb12*-WT mice (*Zhang et al., 2022*). The *Slc26a* family is highly conserved, with 11 members in mice *(SLC26A10* is a pseudogene in humans). Similarly, possible upregulation of two *Slc26a* family members was found in *Slc26a1*-KO mice and base on results from the short-term mRNA suppression, we speculate that it was caused by the long-term loss of *Slc26a1*.

At present, study findings indicate that many genes expressed in the testis are dispensable for mouse fertility (*e.g.*, FBXW17 (*Chen et al., 2022b*), ASB12 (*Zhang et al., 2022*), USP26 (*Felipe-Medina et al., 2019*)). Our results exclude *SLC26A1* as a contraceptive target and male infertility factor and will help avoid duplication of effort by researchers and save time and money in other laboratories. There is only a remote possibility of a germ-cell-specific dominant mutation. These results can also help reproductive researchers determine target gene research priorities and focus on genes that are essential for fertility.

### Funding
This work was supported by the National Key Research and Development Program of China (2022YFC2702702), the University Synergy Innovation Program of Anhui Province (GXXT-2021-071), the Suzhou Science and Technology Development Plan (SZM2021010) and the Introduce Project of Clinical Medicine Experts of Suzhou Industrial Park (SZYQTD202104). The funders had no role in study design, data collection and analysis, decision to publish, or preparation of the manuscript.

## Grant Disclosures

The following grant information was disclosed by the authors:

National Key Research and Development Program of China: 2022YFC2702702.

University Synergy Innovation Program of Anhui Province: GXXT-2021-071.

Suzhou Science and Technology Development Plan: SZM2021010.

Clinical Medicine Experts of Suzhou Industrial Park: SZYQTD202104.

## Competing Interests

The authors declare that they have no competing interests.

## Author Contributions

- Zhixiang Meng performed the experiments, analyzed the data, prepared figures and/or tables, authored or reviewed drafts of the article, and approved the final draft.
- Yu Qiao performed the experiments, analyzed the data, prepared figures and/or tables, and approved the final draft.
- Jiajia Xue performed the experiments, prepared figures and/or tables, and approved the final draft.
- Tiantian Wu performed the experiments, prepared figures and/or tables, animal testing and sample processing, and approved the final draft.
- Wenxin Gao performed the experiments, prepared figures and/or tables, animal testing and sample processing, and approved the final draft.
- Xiaoyan Huang conceived and designed the experiments, authored or reviewed drafts of the article, and approved the final draft.
- Jinxing Lv conceived and designed the experiments, authored or reviewed drafts of the article, and approved the final draft.
- Mingxi Liu conceived and designed the experiments, authored or reviewed drafts of the article, and approved the final draft.
- Cong Shen conceived and designed the experiments, authored or reviewed drafts of the article, and approved the final draft.

## Animal Ethics

The following information was supplied relating to ethical approvals (*i.e.*, approving body and any reference numbers):

The animal study was reviewed and approved by the Animal Ethics and Welfare Committee of Nanjing Medical University (No. IACUC-2004020).

## DNA Deposition

The following information was supplied regarding the deposition of DNA sequences:

The Sanger sequences are available at NCBI: SRR25424720 and SRR25424719.

## Data Availability

Zhixiang Meng. 2023. Slc26a1 is not essential for spermatogenesis and male fertility in mice. figshare. https://doi.org/10.6084/m9.figshare.23635794.v1.

## Supplemental Information

Supplemental information for this article can be found online at http://dx.doi.org/10.7717/peerj.16558#supplemental-information.

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
