# Peer review of "Slc26a1 is not essential for spermatogenesis and male fertility in mice"

_PeerJ, doi:10.7717/peerj.16558_

## Round 0.1 · original submission · Major Revisions

Dear Authors,

I believe your work on the Slc26a1-KO mice phenotype holds potential for publication, but improvements are necessary. Ensure fluent English through professional editing. Enhance references throughout, especially for the fertility claim (line 33). Revise sentences on lines 59-61 for clarity. Figure legends need more detail; distinguishing mice from rats. Conduct further experiments for robust conclusions or adjust existing ones. Integrate references closely with claims. Some figures require improved quality. Please, refer to the reviewers' comments. I encourage you to carefully address the comments and suggestions provided by reviewers. Thoroughly improving your manuscript and your responses to their feedback will increase its chances of acceptance.

**Language Note:** The Academic Editor has identified that the English language must be improved. PeerJ can provide language editing services - please contact us at [email protected] for pricing (be sure to provide your manuscript number and title). Alternatively, you should make your own arrangements to improve the language quality and provide details in your response letter. – PeerJ Staff

·

Basic reporting

No comment

Experimental design

No comment

Validity of the findings

No comment

Additional comments

This manuscript presents compelling evidence supporting the dispensability of Slc26a1 in spermatogenesis and fertility through thorough phenotyping of KO mice. The authors' comprehensive research provides strong support for their conclusive findings.
I have following minor points:
1. Line 60, “…the spermatocyte stage in human testis, spermatocytes, and male germ cells”, it is confused.
2. Line 89, which subline used for C57BL/6, C57BL/6J or C57BL/6N?
3. Line 99, where CD-1 came from?
4. Line 181, which t-test used, unpaired or paired? The statistical method was not described for Fig. 5F, G, I, J, L and M.
5. Line 201, male rats?
6. Fig. 4H, the label on top “TUNNEL” is not correct.
7. Fig. 5B and C, it is essential to label the protein ladder. Please indicate the correct band if one of them is SLC26A1.
8. Scale bar for Fig. 5K was missing.

Reviewer 2 ·

Basic reporting

-English should be edited by a fluent English-speaker or a professional editing company.
-References are inadequate. In the introduction section, almost all sentences need references.

Experimental design

-The names of cell lines should be accurately described: GC-1 spg and GC-2spd(ts).
-In line 98, the authors describe the sequence of guide RNA, but it is actually the DNA sequence because it contains T. How did they prepare sgRNA for injection.
-What are descriptions in lines 101-103? The method of genotyping? If so, the method should be described independently of KO mouse generation.
-The method for fertility test should be described in more details. What did the authors do when one of the two female mice mated with a male got pregnant? Did they remove the pregnant male and put another female?
-In line 143, describe the product number and dilution of the secondary antibody.
-In lines 147-153, the method for RT-PCR should be described in more details. What reverse transcriptase were used? What primers were used for reverse transcription?
-In 160-163, the motility test should be described independently of histology.
-Describe whether the authors showed SEM or SD in each graph.

Validity of the findings

-In lines 187-194, the authors should mention what happened to the SLC26A1 protein by the 10-bp deletion.
-In line 191, it is hard to judge whether the SLC26A1 signal was really in the acrosome partially due to the low magnification of the photo. The authors should immunostain the mature sperm from epididymides.
-I cannot understand the sentence “Slc26a1 affected spermatogenesis via its action on acrosomes” in line 194. Acrosomes function in fertilization, not in spermatogenesis. What mechanism do the authors presume?
-In figure 2H, the authors showed the ratio of abnormal sperm, but there are different types of abnormal sperm morphology. They should describe whether the ratio of each type of abnormal sperm was similar between WT and KO.
-In figure 4D, it seems that Leydig cells were also stained. Is Sox9 expressed in Leydig cells? If so, SOX9 should not be used as a marker for Sertoli cells.
-In lines 237-240, I don’t understand why the authors performed CCK-8 and colony formation assays. How are these results related to the phenotype of Slc26a1-KO mice? They should add the explanation of the reason and the discussion of the interpretation to the manuscript.
-In lines 252-254, it is inappropriate to compare the cell culture model with KO mouse model.

Additional comments

The authors generated Slc26a1-KO mice and investigated its effects on male fertility. They investigated morphology, sperm factors, expression of marker genes, and expression of Slc26 family genes. As a result, they found no significant difference between WT and KO mice and the increase in expression of two Slc26 family genes. They also performed knockdown of Slc26a1 in testicular germ cell lines and observed the decrease in proliferation and colony formation. It is worth reporting the reproductive phenotype of Slc26a1-KO mice even if they were normal. However, there are many points to be considered for publication. In addition to the points described in other sections, I add several more points as follows.

-Because the authors claimed “Scl26a1 was dispensable for mouse fertility” (line 33), they should show the data concerning female fertility.
-The two sentences in lines 59-61 should be re-written. I don’t understand the meaning of them.
-Figure legends should describe a bit more details.
-In line 201, “rats” must be “mice”.
-In line 216, “spermatogenesis” must be “spermiogenesis”.

Reviewer 3 ·

Basic reporting

The submission is written in good English, provides relevant introduction and references, although the citation numbers of references should be given near every claim made. The figures are relevant to the content, with the exception of Fig.1C have a sufficient resolution, and descriptions should be more detailed. The content seems self-contained and coherent.

Experimental design

The paper describes original primary research within Biological Sciences. The manuscript defined relevant and meaningful research question. The knowledge gap was identified, and the study contributes to filling this gap.
The Methods section does not describe the genetic background of the WT controls, the method on sperm staining to analyze malformations, how many sperm cells were analyzed per male, and what malformation classes were counted. Was it counted in solution or on dried spreads? Why was the sperm stained with H-E but not the standard Eosin Y or Coommassie blue? The manuscript must also contain info on how many tubular crossections were analyzed per male for each germ cell marker using histochemistry and for TUNEL, as well as the composition of culture media for CASA, and genetic background of WT controls.
The mutant mice have to registered in the MGI database
https://www.informatics.jax.org/mgihome/submissions/amsp_submission.cgi
and the assigned name has to be used at least once in this manuscript.
The animal research seemed otherwise performed according to ethical standards in the field.

Validity of the findings

Fig. 1C does not allow to conclude that „Slc26a1 was localized in the acrosome of the seminiferous tubules of WT mice“, because there is no such a thing. Dothe authors mean “acrosomes of spermatids” or “acrosome stage of spermiogenesis”? Either way, Fig.1C lacks resolution and marker of acrosome/spermatid, so it does not show any connection to acrosome.
Although most conclusions are based on controlled experiments, the genetic background of WT controls is not specified. Were they outbred? Were the WT controls littermates of the KO mice?
Outbred mice require a larger sample size- this is seen in the high variability in the WT mice. The mean sperm count is lower for KO compared to WT, thus it is necessary to analyze more animals in order to exclude any effect of Slc26a1 deficiency (3 versus 3 is not enough, see also below).
Most of the raw data are provided in the supplement. Body weight and testes weight have to be also reported for individual mice, not just their ratio, as well as their age, because these data can be used to estimate their health and variability. Health status and pathogen policy of the animal facility should be also reported.
RT-qPCR data for Slc26 genes are not significant after adjustment for multiple testing (will be P = 0.09 and 0.08 for Slc26a5 and Slc26a11, resp.), so they cannot be interpreted as conclusive. This could be resolved by adding RT-qPCR data from more animals.
An alternative explanation of the [non-significant] upregulation of two Slc26a family members could be the variable effects of frame-shifting DNA mutation versus mRNA translation suppression. These alternatives could be resolved by engineering a germline miRNA- or siRNA-encoding gene targeting the ORF of Slc26a1 in the testes and RT-qPCR of the Slc26a family members in these testes compared to littermate controls. This issue could be also resolved by mentioning this alternative explanation in the text or changing the claims ("Similarly, upregulation of several Slc26a family members was found in Slc26a1-KO mice and was caused by long-term loss of Slc26a1." is not supported by the results provided).
In Fig.4A, there seems to be less spermatids in the KO compared to WT, but this has not been quantified, despite that spermatids likely express SLC26A1 (Fig1C). The [non-significant] upregulation gene of the two Slc26a family members in total testis RNA can be also alternatively explained by different number of spermatids in KO versus WT testes, so the spermatid quantification is a key missing result.

Additional comments

Additional comments and suggestions [tracked to manuscript] including formatting and word selection are attached as pdf file.

Annotated reviews are not available for download in order to protect the identity of reviewers who chose to remain anonymous.

---

## Round 0.2 · Major Revisions

Dear authors, thank you for your revisions and efforts. The manuscript has been improved, however there have been changes that concern me and the reviewers. Please, refer to their comments for further details.

·

Basic reporting

no comment

Experimental design

no comment

Validity of the findings

no comment

Additional comments

The authors have addressed all my comments and I have no additional comments.

Reviewer 2 ·

Basic reporting

I appreciate authors’ great efforts to revise the manuscript. The revised manuscript is improved very much. However, there is one thing that should be clarified: immunostaining of SOX9 in the revised figure 4E and in the former figure 4D. In the former version, positive strong signals were observed in both Sertoli and Leydig cells, and the authors explained that the staining in Leydig cells is non-specific. If so, why can the authors guarantee that signals within seminiferous tubules are specific? Moreover, in the revised figure 4E, signals in Leydig cells seems to disappear, and those in Sertoli cells are almost invisible. Did the authors make changes in the data during revision? The authors should show clearer signals of SOX9 specifically in Sertoli cells.

Experimental design

no comment

Validity of the findings

no comment

Additional comments

no comment

Reviewer 3 ·

Basic reporting

The authors resolved most of the queries I raised.
However, they introduced a few new problems, fortunately minor ones.

As the authors corrected the info about the genetic background of the mice [not CD-1 but C57BL/6J]: C57BL/6J is actually inbred, so if the authors have used C57BL/6J zygotes and have crossed the mutant mice always to C57BL/6J, "outbred" must be removed from the Abstract [line 31].

The authors answered the question "how many tubular crossections were analyzed per male for each germ cell marker using histochemistry and for TUNEL" (50 tubular crossections were analyzed per male for each germ cell marker using histochemistry and for TUNEL), but only in Cover Letter- this info must be also included in Methods. This section should also contain that “the genetic background of WT controls was C57BL/6J (from the same batch as KO mice, not littermates).”

211 “staining of testes” According to Cover Letter, it is staining of sperm- please resolve.

225 Some sperm is always abnormal, even in the controls.

261 I suggest to add to the text an explanation/justification for doing the cell assays [as also suggested by Reviewer 2]: “Since there was no compensation in these two cell lines, we assessed whether the transient knockdown of Slc26a1 affects their phenotype.”

276-278 It is unclear, in which testicular cell types occurs the overexpression of other Slc26a family members, and this should be specified in the text.

Experimental design

Specific pathogens screened for but absent in their animalhouse should be listed (or refered to a previous publication with this info, if available).

Were the specific sperm malformations [e.g., amorphous head] the same or different from most most widely used reference Wyrobek and Bruce 1975?
https://www.ncbi.nlm.nih.gov/pmc/articles/PMC388734/?page=2

Validity of the findings

209 “lead to frame-shift mutations” must be changed to “is predicted to lead to frame-shift mutation”, because the data on the SLC26A1 protein epitope absence from the KO mice could also be due to instability of protein and/or mRNA.

214 Acrosome localization does not imply effect on acrosome maturation. To show effect on acrosome maturation, several spermatid stages would have to be analyzed.

39-40, “indicated” must be changed to “may indicate”, because
to unequivocally prove the gene compensation, phenotyping results from mice deficient in several Slc26a family genes are necessary. Until then, the compensation is only a possibility.

300-302 I suggest to add to the text “There is only a remote possibility of a germ-cell-specific dominant mutation.”, because the authors examined only the deficiency [permanent null or hypomorphic mutations] of Slc26a1.

Additional comments

I summarized these and some other minor suggestions in the attached file.

Annotated reviews are not available for download in order to protect the identity of reviewers who chose to remain anonymous.

---

## Round 0.3 · accepted · Accept

Dear authors, many thanks for your submission and prompt endeavors. I am happy to let you know that your manuscript has now been accepted for publication.